**Data Availability Statement:** Data Availability: All relevant data are available on the Open Science

# The therapeutic alliance between study participants and intervention facilitators is associated with acute effects and clinical outcomes in a psilocybin-assisted therapy trial for major depressive disorder

Adam W. Levin[1,2], Rafaelle Lancelotta[1], Nathan D. Sepeda[1,3], Natalie Gukasyan[3], Sandeep Nayak[3], Theodore L. Wagener[4,5], Frederick S. Barrett[3], Roland R. Griffiths[3], Alan K. Davis[1,3]*

1 The Ohio State University, Center for Psychedelic Drug Research and Education, Columbus, Ohio, United States of America, 2 Department of Psychiatry, The Ohio State University, Columbus, Ohio, United States of America, 3 Johns Hopkins University, Center for Psychedelic and Consciousness Research, Baltimore, Maryland, United States of America, 4 Center for Tobacco Research, The Ohio State University James Comprehensive Cancer Center, Columbus, Ohio, United States of America, 5 Department of Internal Medicine, The Ohio State University, Columbus, Ohio, United States of America

* davis.5996@osu.edu

## Abstract

We examined if the therapeutic alliance between study participants and intervention facilitators in a psilocybin-assisted therapy (PAT) trial changed over time and whether there were relationships between alliance, acute psilocybin experiences, and depression outcomes. In a randomized, waiting list-controlled clinical trial for major depressive disorder in adults ($N = 24$), participants were randomized to an immediate ($N = 13$) or delayed ($N = 11$) condition with two oral doses of psilocybin (20mg/70kg and 30mg/70kg). Ratings of therapeutic alliance significantly increased from the final preparation session to one-week post-intervention ($p = .03$, $d = .43$). A stronger total alliance at the final preparation session predicted depression scores at 4 weeks ($r = -.65$, $p = .002$), 6 months ($r = -.47$, $p = .036$), and 12 months ($r = -.54$, $p = .014$) post-intervention. A stronger total alliance in the final preparation session was correlated with higher peak ratings of mystical experiences ($r = .49$, $p = .027$) and psychological insight ($r = .52$, $p = .040$), and peak ratings of mystical experience and psychological insight were correlated with depression scores at 4 weeks ($r = -.45$, $p = .030$ for mystical; $r = -.75$, $p < .001$ for insight). Stronger total alliance one week after the final psilocybin session predicted depression scores at 4 weeks ($r = -.85$, $p < .001$), 3 months ($r = -.52$, $p = .010$), 6 months ($r = -.77$, $p < .001$), and 12 months ($r = -.61$, $p = .001$) post-intervention. These findings highlight the importance of the therapeutic relationship in PAT. Future research should explore therapist and participant characteristics which maximize the therapeutic alliance and evaluate its relationship to treatment outcomes.

**Trial registration:** Registration: Clinicaltrials.gov NCT03181529. https://classic.clinicaltrials.gov/ct2/show/NCT03181529.

Framework public repository at https://osf.io/speqc/?view_only=90add9a193b04fa9ba3c5865c3bc15fb Identifier: DOI 10.17605/OSF.IO/SPEQC.

**Funding:** This study was funded in part by a crowd-sourced funding campaign organized by Tim Ferriss and a grant from the Riverstyx Foundation. AKD, NDA, NG, SN, FSB, and RRG are supported by funding from Tim Ferriss, Matt Mullenweg, Craig Nerenberg, Blake Mycoskie, the Steven and Alexandra Cohen Foundation. AKD, RL, and AWL are supported by the Center for Psychedelic Drug Research and Education in the College of Social Work at Ohio State University, funded by anonymous private donors. The funder had no role in study design, data collection and analysis, decision to publish, or preparation of the manuscript. AKD and RL are board members of Source Research Foundation. AKD is a Lead Trainer at Fluence. FB is a scientific advisor for WavePaths, Ltd and Mindstate Design Labs, Inc, and has provided consultation services for Gilgamesh Pharmaceuticals, Inc. These organizations were not involved in the design/execution of this study or the interpretation or communication of findings.

**Competing interests:** AKD and RL are on the board of Source Research Foundation. RRG is on the board of the Heffter Research Institute. This does not alter our adherence to PLOS ONE policies on sharing data and materials.

## Introduction

Psychedelic-assisted therapy (PAT) is a therapeutic intervention which has shown promise in the treatment of a wide range of psychiatric disorders, including depression [1–4], post-traumatic stress disorder (PTSD) [5], addiction [6,7], and end-of-life anxiety and depression [8]. Although this approach centers around the administration of a psychedelic drug, it has long been claimed that the psychological milieu, as well as the physical environment of the drug administration, referred to as the 'set' and 'setting' respectively, are important in determining the quality of the acute psychedelic experience [9–11]. One aspect of the 'set and setting', emphasized in modern clinical trials, are the hours of preparatory and integrative psychotherapy surrounding the drug administration, as well as the supportive, nondirective, therapy during the actual drug session [12–14]. Although some have suggested the superiority of specific therapeutic orientations [15], thus far, a range of therapeutic approaches have produced comparable outcomes (e.g. cognitive behavioral therapy [16], motivational enhancement therapy [6], acceptance and commitment therapy (e.g., Accept-Connect-Embody [2]). This has prompted others to advocate for a 'common factors' framework, which emphasizes the common elements between therapeutic approaches rather than those associated with any specific orientation [17].

One of the most widely investigated of these 'common factors' is the therapeutic alliance, broadly defined as the sense of collaboration between patient and clinician [18–20]. Decades of research show that a stronger alliance is associated with positive treatment outcomes, and this association appears to be robust and consistent across disorders, cultures, and theoretical orientations (for recent reviews see: [21,22]). To date, few studies have empirically investigated the effect of the therapeutic alliance on outcomes in PAT. Some observational research has suggested that various states and traits traditionally associated with a strong alliance, such as acceptance, openness to experience, and a strong intention are associated with better therapeutic outcomes [23,24]. However, this evidence is indirect and based on psychedelic use in naturalistic settings where a traditional therapeutic alliance may not be established.

In a recent study of PAT, Murphy et al. (2022) [25] found that the strength of the therapeutic alliance predicted depression scores at follow-up and that this effect was mediated by the intensity of the acute psychedelic experience. Specifically, they demonstrated that a strong therapeutic alliance predicted a positive pre-psychedelic session rapport, which then predicted greater emotional breakthrough and mystical-type experiences during the psilocybin session leading to improved therapeutic outcomes [25]. Importantly, their analysis also demonstrated that a weaker alliance predicted less emotional breakthrough and mystical quality of the acute experience.

The mystical-type experience, as measured by the Mystical Experience Questionnaire (MEQ), has been, to date, the most extensively researched dimension of the acute psychedelic experience and has been correlated with positive therapeutic outcomes in multiple clinical trials and across diagnoses [26,27]. However, recent research suggests that psychological insight, especially when paired with a mystical experience, might be a more robust predictor of change [28]. Although many psychotherapeutic approaches, especially in the psychodynamic tradition, emphasize the emergence of insight in the context of a strong therapeutic alliance [20,29,30], this relationship has yet to be empirically investigated in the context of PAT. Additionally, although Murphy et al.'s analysis suggests that a weaker alliance might limit therapeutic effectiveness, the causal mechanisms of this are unknown and have not been investigated.

Therefore, in a replication and extension of Murphy et al. (2022) [25], we analyzed data from a recently published randomized, waitlist-controlled clinical trial of psilocybin-assisted therapy for major depressive disorder (MDD) [3] to examine: 1) whether the participant-rated

therapeutic alliance varies from before to after psilocybin administration sessions, 2) whether the alliance is correlated with depression scores following the intervention, 3) whether the therapeutic alliance is correlated with peak mystical-type and psychological insight effects of the intervention, and 4) the relationship between ratings of the mystical-type and psychological insight effects and post-treatment depression scores.

# Methods

## Study design

The primary outcomes from this study have been published and full details, including a study protocol, have been described previously [3]. Procedures were approved by The Johns Hopkins University School of Medicine Institutional Review Board (IRB00101821). In brief, this study was a randomized waitlist-controlled trial testing the effect of two ascending doses of psilocybin in conjunction with approximately 11 hours of psychotherapy in adults with moderate-to-severe MDD. The decision to recruit 24 participants was made based on the results of a prior study of psilocybin-assisted therapy for cancer patients [8], which showed very large effect sizes of this intervention. Therefore, as a preliminary trial we were confident that a sample size of $N = 24$ would provide sufficient power to detect a moderate effect size of pre-psilocybin to post-psilocybin change in depressive symptoms. Recruitment occurred between August 2017 to August 2019 and all participants provided written informed consent. Data were collected through final 12-month follow-up timepoint in August 2020. Participants in this trial were 21–75 years of age who met criteria for a moderate to severe episode of MDD ($\geq$17 on the GRID-Hamilton Depression Rating Scale [GRID-HAMD]) and were medically stable. Individuals with a first- or second-degree relative with history of psychiatric conditions (e.g., psychotic or bipolar I or II disorder) deemed incompatible with safe exposure to psilocybin were excluded. Participants were also required to refrain from using certain medications (e.g., antidepressants) for at least five half-lives before screening and through our primary outcome measurement at 4-weeks post psilocybin session two. Following screening and baseline assessments, participants were randomized to an immediate or delayed treatment condition.

During the intervention phase of the study, participants were provided with approximately 8 hours of preparatory therapy sessions with two intervention facilitators. At least one facilitator had clinical training in mental health at the master's or doctoral level (e.g., Master of Social Work, Ph.D. in clinical psychology, M.D. specializing in psychiatry). Following preparation sessions, participants received two doses (20 mg/70 kg and 30 mg/70 kg) of psilocybin, approximately one-to-two weeks apart. Psilocybin was administered under the supervision of both facilitators in a comfortable room following established safety guidelines [12]. On psilocybin session days, a nondirective psychotherapeutic approach was employed. The following day and one week after each psilocybin session, participants returned for follow-up integrative therapy sessions and assessments. Follow-up assessments, including a 1–2 hour meeting with at least one of the therapist facilitators, were also conducted at 1- and 4-weeks, and 3-, 6-, and 12-months following the second psilocybin session.

## Measures

**Therapeutic alliance.** The 12-item, participant-rated, Working Alliance Inventory-Short Revised (WAI-SR; [31]) was used to assess the therapeutic alliance between intervention facilitators and study participants at the final preparatory session prior to psilocybin administration and one week following the first and second psilocybin sessions. The WAI-SR is a shortened version of the 36-item Working Alliance Inventory (WAI; [32,33]) which utilizes a three factor model based on the three aspects of the therapeutic alliance as theorized by Bordin [18]: The

goals of treatment (e.g., "____ and I collaborate on setting goals for my therapy"), the tasks of therapy (e.g., "As a result of these sessions I am clearer as to how I might be able to change"), and the bond between therapist and client (e.g., "I believe ____ likes me"). The 12 items are rated on a five-point Likert scale ranging from *Seldom* (1) to *Always* (5). Internal consistency reliability, measured with Cronbach's alpha, was calculated for WAI bond ($\alpha$ = .61, .77, .94), goal ($\alpha$ = .89, .91, .95), task ($\alpha$ = .76, .92, .96), and total score ($\alpha$ = .87, .92, .95) at the final preparatory session, 1-week post-psilocybin session 1, and 1-week post-psilocybin session 2, respectively.

**Depression severity.**   Depression was measured by blinded clinician rating using the GRID-HAMD [34,35], as previously described [3]. Depression severity was assessed at baseline and at 1-, 3-, 6-, and 12-month follow-up timepoints. Inter-rater reliability was 87.5% across all follow-ups.

**Acute psilocybin effects.**   Measures of acute psilocybin effects that were assessed at the end of the drug administration or the following day have been reported previously [3]. Included in this follow-up analysis is the Mystical Experience Questionnaire (MEQ30) [36], which was rated on a 6-point scale ranging from 0–5 (none; not at all = 0, so slight cannot decide = 1, slight = 2, moderate = 3, strong [equivalent in degree to any other strong experience] = 4, extreme [more than any other time in my life and stronger than 4] = 5). We also included a single item measure of psychological insight [37] on which participants rated the degree to which the psilocybin session was psychologically insightful on a scale from 1 = "no more than routine, everyday experiences" to 8 = "the single most psychologically insightful experience of my life." The use of a single-item measure, while not ideal, has shown reliability and validity in similar psychological interventions [38].

## Statistical analyses

Data were analyzed from all 24 participants who completed the full intervention. Four participants did not complete the WAI-SR during the final preparatory session and those data were therefore excluded. Additionally, four participants did not complete the psychological insight item, so those data were excluded for the relevant analyses. First, we explored the distribution of scores for each variable and found that several variables violated the assumption of a normal distribution. However, given that Pearson correlations and t-tests are robust against violations of the normal distribution assumption [39–41], and given that this study is exploratory and hypothesis-generating, we chose to proceed with our analytic plan. Next, we used paired samples *t* tests to compare alliance scores from the final preparatory session to one week following the second psilocybin session. Following this, we calculated Pearson's correlations to examine the relationship between participant ratings of the alliance at the final preparatory session and one week following the first and second psilocybin session, and depression scores at 4-week, and 3-, 6-, and 12-month follow ups. Then, we calculated Pearson's correlations between alliance ratings at the final preparatory session and one week following the first and second psilocybin session, and measures of peak acute mystical and insight effects across the two psilocybin sessions. Finally, we calculated Pearson's correlations between peak acute effects and depression scores at 4-weeks, 3-, 6-, and 12-month follow ups. A *p*-value of .05 was used to determine statistical significance in all analyses, and effect sizes are reported for all t-test and correlations. Data analysis was conducted using SPSS, version 28 [42].

## Results

The sample (n = 20) was comprised of 15 women (75%) and 5 men (25%) and was 90% white, with a mean (*SD*) age of 37.6 (11.0) years. Most participants also reported they were

**Table 1. Comparisons of therapeutic alliance ratings (via the working alliance inventory- short revised) at intervention time-points.**

| WAI-SR | Preparatory Session 2 | | 1-week post-psilocybin session 2 | | | | | |
|---|---|---|---|---|---|---|---|---|
| | M | SD | M | SD | T | p | | Cohen's d |
| Bond | 18.55 | 1.70 | 19.29 | 2.12 | 1.56 | .135 | | .34 |
| Goal | 17.40 | 3.28 | 17.79 | 3.56 | .50 | .620 | | .11 |
| Task | 14.85 | 2.80 | 17.17 | 3.70 | 3.99 | < .001*** | | .65 |
| Total | 50.80 | 6.55 | 54.25 | 8.27 | 2.40 | .027* | | .43 |

*$p < .05$,

**$p < .01$,

***$p < .001$.

$d = .2$ (small), $d = .5$ (medium), $d = .8$ (large).

heterosexual (85%), non-Hispanic (100%), had a college degree (90%), were never married (55%), and employed (55%). On average, the sample reported it had been 20.7 ($SD = 18.5$) months since their current major depressive episode began, and 19.6 ($SD = 11.4$) years since their first major depressive episode.

As Table 1 shows, participant ratings of the total therapeutic alliance between study participants and intervention facilitators increased from the final preparatory session to 1-week post-intervention ($p = .027$, $d = .43$). Although each of the subscales increased during this time point ($p = .620$, $d = .11$ for goal, $p = .135$, $d = .34$ for bond), only ratings of task increased significantly ($p < .001$, $d = .65$).

As Table 2 shows, a stronger total alliance reported at the final preparatory session predicted depression scores at 4 weeks ($r = -.65$, $p = .002$), 6 months ($r = -.47$, $p = .036$), and 12 months ($r = -.54$, $p = .014$) post-intervention. A stronger total alliance reported 1-week

**Table 2. Correlation between therapeutic alliance ratings (via the working alliance inventory- short revised) and depression outcomes (GRID-HAMD).**

| WAI | | GRIDHAMD (4-weeks post-session 2) | GRIDHAMD (3-month follow-up) | GRIDHAMD (6-month follow-up) | GRIDHAMD (12-month follow-up) |
|---|---|---|---|---|---|
| Bond | Final Preparatory Session (N = 20) | -.454* | -.062 | -.228 | -.355 |
| | 1-week post-psilocybin session 1 | -.502* | -.375 | -.470* | -.524** |
| | 1-week post-psilocybin session 2 | -.674*** | -.571** | -.680*** | -.718*** |
| Goal | Final Prep | -.491* | -.229 | -.362 | -.489* |
| | 1-week post-session 1 | -.539** | -.274 | -.541** | -.417* |
| | 1-week post-session 2 | -.688*** | -.290 | -.594** | -.386 |
| Task | Final Prep | -.673** | -.353 | -.540* | -.480* |
| | 1-week post-session 1 | -.783*** | -.448* | -.745*** | -.502* |
| | 1-week post-session 2 | -.853*** | -.551** | -.768*** | -.589** |
| Total | Final Prep | -.651** | -.282 | -.471* | -.542* |
| | 1-week post-session 1 | -.734*** | -.420* | -.710*** | -.541** |
| | 1-week post-session 2 | -.851*** | -.518** | -.774** | -.614** |

*$p < .05$,

**$p < .01$,

***$p < .001$.

$r = .1$ (small), $r = .3$ (medium), $r = .5$ (large).

**Table 3. Correlation between therapeutic alliance (via the working alliance inventory- short revised) and peak acute mystical-type and insight effects.**

| WAI | | Mystical Experiences Questionnaire (MEQ) *N* = 20 | Psychological Insight Item *N* = 16 |
|---|---|---|---|
| **Bond** | Final Preparatory Session | .487* | .467 |
| | 1-week post-session 1 | .416* | .648** |
| | 1-week post- session 2 | .447* | .814*** |
| **Goal** | Final Prep | .237 | .359 |
| | 1-week post-session 1 | .17 | .481* |
| | 1-week post-session 2 | .155 | .596**** |
| **Task** | Final Prep | .579** | .531* |
| | 1-week post-session 1 | .469* | .812*** |
| | 1-week post-session 2 | .439* | .831*** |
| **Total** | Final Prep | .493* | .518* |
| | 1-week post-session 1 | .392 | .746*** |
| | 1-week post-session 2 | .378 | .826*** |

*$p < .05$,

**$p < .01$,

***$p < .001$.

$r = .1$ (small), $r = .3$ (medium), $r = .5$ (large).

following the first psilocybin session was also correlated with decreased depression scores at 4 weeks ($r = -.73$, $p < .001$), 3 months ($r = .42$, $p = .041$), 6 months ($r = -.71$, $p < .001$) and 12 months ($r = -.54$, $p = .006$). A stronger total alliance reported 1-week after the second psilocybin session predicted depression scores at 4 weeks ($r = -.85$, $p < .001$), 3 months ($r = -.52$, $p = .010$), 6 months ($r = -.77$, $p < .001$), and 12 months ($r = -.61$, $p = .001$).

As Table 3 shows, a stronger total therapeutic alliance reported in the final preparation session was correlated with higher peak ratings of mystical-type experiences ($r = .49$, $p = .027$) and of psychological insight ($r = .52$, $p = .040$). A stronger therapeutic alliance reported 1-week following the first psilocybin session was significantly correlated with higher peak ratings of psychological insight ($r = .75$, $p < .001$) across both psilocybin sessions. Higher peak ratings of psychological insight across both psilocybin sessions significantly predicted higher total alliance ratings reported 1-week following the final psilocybin session ($r = .83$, $p < .001$), and higher peak ratings of mystical-type experiences and psychological insight predicted higher bond ratings reported one week following the second psilocybin session ($r = .45$, $p = .029$ for mystical-type, $r = .81$, $p < .001$ for insights).

As Table 4 shows, higher peak ratings of mystical-type experience were correlated with lower depression scores at 4 weeks ($r = -.45$, $p = .030$), and higher peak ratings of psychological insight were correlated with lower depression scores at 4 weeks ($r = -.75$, $p < .001$), 3 months ($r = -.52$, $p = .020$), 6 months ($r = -.82$, $p < .001$), and 12 months ($r = -.70$, $p < .001$) post-intervention.

Our analysis revealed similar patterns with regards to the three WAI subscales. Correlations between the task alliance subscale and outcomes were most robust, with higher ratings of task alliance at the final preparatory session significantly predicting depression scores at 4 weeks ($r = -.67$, $p = .001$), 6 months ($r = -.54$, $p = .014$) and 12 months ($r = -.48$, $p = .032$) post-intervention. Further, ratings of task alliance 1-week following the second psilocybin session were strongly correlated with decreased depression scores at 4 weeks ($r = -.85$, $p < .001$), 3 months ($r = -.55$, $p = .005$), 6 months ($r = -.77$ $p < .001$), and 12 months ($r = -.59$, $p = .013$) months.

**Table 4. Correlation between peak acute mystical-type and insight effects and depression outcomes (GRID-HAMD).**

| Peak Acute Effects | 4-weeks post-session 2 | 3-month follow-up | 6-month follow-up | 12-month follow-up |
|---|---|---|---|---|
| Mystical Experiences Questionnaire (MEQ) $N = 24$ | -.445* | -.156 | -.291 | -.307 |
| Psychological Insight Item $N = 20$ | -.748*** | -.515* | -.819*** | -.704*** |

*$p < .05$,
**$p < .01$,
***$p < .001$.
$r = .1$ (small), $r = .3$ (medium), $r = .5$ (large).

Higher ratings of goal alliance at the final preparatory session predicted outcomes at 4 weeks ($r = -.49$, $p = .028$) and 12 months ($r = -.49$, $p = .029$), and ratings of goal alliance 1-week following the second psilocybin session were correlated with outcomes at 4 weeks ($r = -.69$, $p < .001$) and 6 months ($r = -.59$, $p = .002$). A stronger bond reported at the final preparation session predicted scores at 4 weeks ($r = -.45$, $p = .044$) but did not predict long term depression scores ($ps > .05$). However, a stronger bond reported 1-week after the final psilocybin session predicted depression scores at 4-weeks ($r = -.67$, $p < .001$), 3-months ($r = -.57$, $p = .004$), 6 months ($r = -.68$, $p < .001$), and 12 months ($r = -.72$, $p < .001$).

With regards to the WAI subscales relationship with acute psilocybin effects, ratings of task alliance at the final preparatory session were correlated with higher ratings of both mystical-type experience ($r = .58$, $p = .007$) and psychological insight ($r = .53$, $p = .034$). Further, higher peak ratings of both mystical and psychological insight predicted higher task alliance ratings 1-week following the second psilocybin session ($r = .44$, $p = .032$ for mystical-type, $r = -.83$, p $< .001$ for insight). A stronger bond reported in the final preparation session was significantly correlated with higher peak ratings of mystical-type experiences ($r = .49$, $p = .029$). Additionally, a stronger bond reported one week following the second psilocybin session was significantly correlated with mystical-type experiences ($r = .45$, $p = .029$) and psychological insight ($r = .81$, $p < .001$). Ratings of goal alliance at the final preparatory session were not significantly correlated with peak ratings of mystical-type experience ($r = .24$, $ps > .05$) or psychological insight ($r = .36$, $ps > .05$), however, peak ratings of psychological insight predicted goal ratings one week following the second psilocybin session ($r = .60$ $p = .006$).

## Discussion

This analysis of a randomized controlled trial of psilocybin-assisted therapy for adults with MDD found that a stronger participant-rated therapeutic alliance was correlated with improvements in depression outcomes up to one year later, and that this correlation strengthened throughout the course of the study. A stronger alliance in the final preparatory session predicted higher peak ratings of mystical-type and psychological insight experiences, and these acute effects were correlated with improvements in depression. These results suggest that therapeutic alliance plays an important role in optimizing the acute psychedelic experience towards improved outcomes and extend those of Murphy et al. (2022) [25] by employing clinician-rated depression outcomes (vs self-report), including acute psychological insight and mystical-type effects (vs emotional breakthrough and mystical-type effects), and extending the timeline to 12-months post-intervention (vs 6-weeks).

In one sense, these findings place PAT firmly within the bounds of traditional psychotherapies, in which a stronger therapeutic alliance is correlated with positive outcomes across modalities and diagnoses [21,22,43,44]. In another sense, the alliance seems to have an

amplified role in PAT through its interaction with the acute psychedelic experience. Several large meta-analyses of traditional therapies have placed the alliance-outcome correlation consistently in the small to medium range [22,45–47], and a recent meta-analysis, looking specifically at the alliance-depression outcome correlation, found a small effect [48]. By contrast, our analysis showed moderate to large correlations. It is possible that outliers in our sample, or positive expectancy effects, drove these correlations [2,49], however, our results suggest other possible explanations. For example, the task alliance subscale is often associated with improved outcomes in psychotherapy, but our finding that the bond subscale correlated with outcomes is significantly less common [50–55]. If psychedelics are 'unspecific amplifiers of mental processes', as some have suggested [56], then the participant/facilitator bond itself may be subject to that amplification, leading to larger effects.

This model supports many long-held assumptions, in both the traditional and contemporary use of psychedelics, about the importance of the therapist or facilitator in a therapeutic psychedelic experience. Traditional Ayahuasca practitioners undergo extensive apprenticeships and are thought to directly affect healing through emotional and spiritual 'resonance' during complex ceremonies [57]. Modern clinical trials employ a therapeutic dyad with two session facilitators per participant present throughout dosing sessions and during extensive preparatory and integrative therapeutic work. Safety guidelines include recommendations for ancillary staff given that "all individuals at the study site having contact with the volunteer on or before the session day may influence a volunteer's reaction to a hallucinogen" [11, p.12].

Although assumptions about the importance of facilitators through altered states of consciousness are pervasive, they have only recently been empirically substantiated. A large survey of ayahuasca users in ceremonial settings showed that 'adequate leadership' was associated with higher levels of mystical-type experience, and that clear instructions and feelings of social adhesion predicted fewer challenging experiences [58]. Other naturalistic survey studies have demonstrated a correlation between 'clear intentions' and mystical-type experiences [23,24], and a recent prospective survey study of participants at psychedelic retreats found that rapport with facilitators prior to a group psychedelic session mediated a sense of harmony during the session [59]. However, the present study and that of Murphy et al. [25] are the first to demonstrate a correlation between the contemporary construct of the therapeutic alliance and the quality of the acute psychedelic experience. Within this construct, it seems plausible that the task alliance subscale of the WAI, which was most robustly correlated with outcomes in our analysis, might be analogous to the concept of 'clear intentions' or 'adequate leadership', within the naturalistic models outlined above, while the bond subscale may more accurately reflect the rapport or 'communitas' [59]. However, further research is needed to elucidate the overlap and contrasts between these constructs.

The correlation between the strength of the acute psychedelic experience and clinical outcomes has been well established in the modern era of psychedelic research [60] and was further validated in the present study. The most widely used construct of the subjective experience (e.g., mystical-type) has been correlated with positive therapeutic outcomes in multiple clinic trials and across indications [26,27]. Interestingly, in both ours and Murphy et al.'s (2022) analyses, mystical-type scores were less predictive of improvements in depression than were other measures of acute effects. In our analysis, although both peak ratings of psychological insight and mystical-type experience correlated with decreases in depression at 4 weeks, only ratings of psychological insight correlated with depression scores at long-term follow-up. Similarly, Murphy et al. (2022) found that ratings of emotional breakthrough were more predictive of decreases in depression than ratings of the mystical-type experience and that emotional breakthrough was more predictive in the first psilocybin session while mystical-type experiences were more predictive in the second session.

These findings suggest that the combination of a more psychologically grounded experience with a mystical-type experience might be a more robust predictor of long-term therapeutic change than the mystical-type experience alone [28,61,62] and lend support to a model in which working through interpersonal content occurs prior to mystical-type breakthroughs [56,62,63]. If future research confirms these findings in more diverse samples and in other clinical indications, it suggests that PAT might be best viewed as an inter- and intra-personal process, which can be enhanced via a strong alliance between study participants and intervention facilitators.

Our findings also strongly suggest that the acute psychedelic experience directly enhances the therapeutic alliance. Psilocybin's ability to enhance emotional empathy [64] and feelings of connectedness [59,65], along with its reduction of subjective and neurological responses to social exclusion [66], might explain this phenomenon. These effects not only persist well beyond the acute drug effect [67] but may also foster more secure attachment patterns [68]. In our study, stronger alliances one week after the second psilocybin session correlated with higher peak ratings of acute effects. Murphy et al., (2022) also found that stronger emotional breakthroughs in the first psilocybin session predicted better alliances before the second session. Collectively, these results suggests that psilocybin could directly and progressively enhance the therapeutic alliance.

The effects of the alliance on depression outcomes increased over the course of the intervention, with the strongest effects following the second psilocybin session. In Murphy et al. (2022), ratings of alliance reported prior to the second psilocybin session, but not prior to the first, predicted depression outcomes in a manner that was *independent* of the acute psychedelic experience in the second session. Similarly, a recent large, prospective survey study of psychedelic use in guided group retreat settings showed that the enduring positive effects were explained by the extension of feelings of social connectedness and bond *beyond the acute psychedelic state* [59]. Given that the therapeutic alliance itself is likely an independent driver of therapeutic change [21], this suggests that positive therapeutic outcomes in PAT may be in part driven by the enhancement and extension of the alliance beyond the acute psychedelic experience. Thus, PAT might be best viewed as a progressive and reciprocal process, as opposed to a one-off, short-term, intervention. Namely, the psilocybin sessions are both enhanced by, and enhance, the therapeutic alliance towards improved therapeutic outcomes over time.

This model has many implications for both treatment approaches and future areas of research. For one, the dynamic and reciprocal nature of the alliance would seem to recommend both flexible dosing and timing of dosing sessions, as well as frequent assessment and repair of alliance ruptures [25,43,44]. There may be specific doses, or even specific compounds, which are more suited for certain stages of therapy, for example, in strengthening the therapeutic alliance or in working through interpersonal, versus spiritual or existential, content [69]. Such approaches have been employed historically, for example, in the 'psycholytic' model, which used lower doses of psychedelics to elicit and work through interpersonal and psychodynamic content, but are underrepresented in contemporary research [62]. Recently, in a real-world clinical study of psychedelic-assisted group therapy, most participants required multiple dosing sessions (ranging from 1 to 12 per participant), where MDMA was used preferentially and initially to establish the therapeutic alliance, and higher doses of classic psychedelics were used only subsequently [63]. Notably, the primary diagnosis in this study was complex-PTSD, and all participants were already in regular psychotherapy with the authors. This approach, while preliminary, may presage the realities inherent to the real-world implementation of PAT, especially in the context of certain complex, chronic diagnoses.

Perhaps the clearest implication of our findings is the potential benefits of the long-term maintenance of the therapeutic alliance beyond the timeline of typical clinical trials. Although this potential has long been recognized in traditional and historical approaches to work with psychedelics, the incentives associated with commercialization and medicalization may attempt to reframe PAT as a medicine administration procedure and to undermine the importance of therapeutic support [70]. It will be important to continue to investigate and empirically validate the role of the therapeutic alliance and to attempt to develop cost-effective approaches that incorporate these findings.

Future research should define and measure the components of a strong therapeutic alliance in the context of PAT. Although there will undoubtedly be significant overlap with traditional therapeutic modalities, it is likely that some contingencies unique to PAT will require novel analytic approaches and training [71]. Much of the therapeutic support provided in psychedelic sessions is non-verbal [72] and 'supportive' or 'therapeutic' touch is commonly employed [12,73]. While physical touch is often discouraged in traditional psychotherapy, evidence suggests that it may be essential in providing support and connection in psychedelic states [72]. It is therefore important that mutual agreement and informed consent are directly addressed and defined. Additionally, there is the question of whether therapists should themselves have personal experience with psychedelics in order to provide clear guidance and to support informed consent. Although such personal training experiences are considered essential in indigenous traditions [57], as well as to other modalities of psychotherapy, such as psychoanalysis [20,30], current laws in the United States prohibit psychedelic use for training purposes. As a result, little is known empirically about how personal psychedelic experience may or may not impact the therapeutic alliance, though there are some indications that participants might prefer facilitators with personal experience [74]. In the coming years, as training opportunities with psychedelics become more widely and legally available [75], it will be important to evaluate this relationship empirically (for an in depth review of this topic, see: [76]).

## Limitations

There are several limitations of this study. This study was conducted prior to the development of several psychological insight scales validated for use in PAT [77,78]. As such, we used a single item measure for insight, which may limit the results. However, it is worth noting that such measures have demonstrated excellent test-retest reliability and validity in other similar studies and may be beneficial in limiting participant burden. Our sample consisted of individuals who self-referred for a novel treatment intervention which has recently received significant positive media attention. As a result, participants may have been more susceptible to expectancy bias and the placebo effect [49], and may have been more likely or willing to form strong, trusting bonds with their intervention facilitators, thus exaggerating the role of the therapeutic alliance in predicting psilocybin effects and outcomes. In addition, we were not able to control for any participant, therapist, or treatment setting influences, perhaps most impacted by the fact that two therapy facilitators are incorporated in this intervention. Future studies could explore whether the alliance with one, or both, facilitators impacts the relationships with acute effects and therapeutic outcomes. Future studies should include multiple raters of alliance (including third-party) to elucidate the dynamic relations among the triad and its impact on outcomes [79].

Because most participants were White (92%), Female (67%), and heterosexual (96%), we also caution against extrapolating our findings to more diverse populations, as it is possible that this participant demographic would have a higher likelihood of forming strong alliances with intervention facilitators [80]. Additionally, we were unable to control for diversity in

therapist demographics or their orientations and approaches to providing therapy which could have played a role in explaining improvements related to alliance, acute psilocybin effects, and depression outcomes. Further studies should address the influence of these factors on alliance formation by ensuring more diverse participant populations and discussing the identities of the intervention facilitators. Lastly, we also found that there were violations of the assumption of normality in our data. Although evidence shows that t tests and Pearson correlations are robust against such violations, we nevertheless suggest that these data be interpreted with caution and that future studies with larger samples replicate these findings.

## Conclusion

The strength of the participant-rated therapeutic alliance was positively correlated with decreases in depression up to one-year following a psilocybin-assisted therapy intervention for depression. The correlation strengthened throughout the course of the intervention and was strongest at the final integration session. The strength of the therapeutic alliance prior to, and following, psilocybin dosing sessions was correlated with peak ratings of mystical-type experience and psychological insight, and these ratings were correlated with depression scores at follow-up. These findings reveal a reciprocal relationship between the therapeutic alliance and the acute psychedelic experience, in which each acts to strengthen the other over the course of an intervention towards improved outcomes. Psilocybin may thus be best administered in the context of a long-term therapeutic relationship. Future research should seek to replicate these findings in different therapeutic settings and approaches, and to elucidate the characteristics, of both participants and therapists, which facilitate an optimal therapeutic alliance.

## Author Contributions

**Conceptualization:** Adam W. Levin, Rafaelle Lancelotta, Nathan D. Sepeda, Frederick S. Barrett, Roland R. Griffiths, Alan K. Davis.

**Data curation:** Nathan D. Sepeda, Frederick S. Barrett, Roland R. Griffiths, Alan K. Davis.

**Formal analysis:** Adam W. Levin, Rafaelle Lancelotta, Nathan D. Sepeda, Sandeep Nayak, Frederick S. Barrett, Roland R. Griffiths, Alan K. Davis.

**Funding acquisition:** Frederick S. Barrett, Roland R. Griffiths, Alan K. Davis.

**Investigation:** Frederick S. Barrett, Roland R. Griffiths, Alan K. Davis.

**Methodology:** Frederick S. Barrett, Roland R. Griffiths, Alan K. Davis.

**Project administration:** Adam W. Levin, Nathan D. Sepeda, Natalie Gukasyan, Frederick S. Barrett, Roland R. Griffiths, Alan K. Davis.

**Resources:** Nathan D. Sepeda, Frederick S. Barrett, Roland R. Griffiths, Alan K. Davis.

**Software:** Nathan D. Sepeda, Frederick S. Barrett, Roland R. Griffiths, Alan K. Davis.

**Supervision:** Natalie Gukasyan, Frederick S. Barrett, Roland R. Griffiths, Alan K. Davis.

**Validation:** Nathan D. Sepeda, Frederick S. Barrett, Roland R. Griffiths, Alan K. Davis.

**Visualization:** Adam W. Levin, Nathan D. Sepeda, Frederick S. Barrett, Roland R. Griffiths, Alan K. Davis.

**Writing – original draft:** Adam W. Levin, Rafaelle Lancelotta, Nathan D. Sepeda, Natalie Gukasyan, Sandeep Nayak, Frederick S. Barrett, Roland R. Griffiths, Alan K. Davis.

**Writing – review & editing:** Adam W. Levin, Rafaelle Lancelotta, Nathan D. Sepeda, Natalie
Gukasyan, Sandeep Nayak, Theodore L. Wagener, Frederick S. Barrett, Roland R. Griffiths,
Alan K. Davis.

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
