## [Decision Letter · Decision Letter 0]

30 Nov 2023

PONE-D-23-21083The therapeutic alliance between study participants and intervention facilitators is associated with acute effects and clinical outcomes in a psilocybin-assisted therapy trial for Major Depressive DisorderPLOS ONE

Dear Dr. Davis,

Thank you for submitting your manuscript to PLOS ONE. After careful consideration, we feel that it has merit but does not fully meet PLOS ONE’s publication criteria as it currently stands. Therefore, we invite you to submit a revised version of the manuscript that addresses the points raised during the review process.

 Three reviewers have made helpful comments to increase the reach of your paper at PLoS One. Reviewer One, in particular, has provided an excellent and detailed list of comments. Reviewer Three captures the essence of comments made by Reviewer Two, and thus addresses those primary concerns about the sample size and how to approach the data analyses sufficiently. Therefore, when resubmitting please outline your responses and edits according to only Reviewer's 1 & 3.

We look forward to receiving your revised manuscript.

Kind regards,

Herb Covington, Ph.D.

Academic Editor

PLOS ONE

Additional Comments from Editor: Please note that I have acted as a reviewer for this manuscript, and you will find my comments below, under Reviewer 3.

“AKD and RL are on the board of Source Research Foundation. RRG is on the board of the Heffter Research Institute. AKD is a lead trainer at Fluence.”

Reviewers' comments:

Reviewer's Responses to Questions

**Comments to the Author**

1. Is the manuscript technically sound, and do the data support the conclusions?

Reviewer #1: Yes

Reviewer #2: Yes

Reviewer #3: Yes

2. Has the statistical analysis been performed appropriately and rigorously? 

Reviewer #1: No

Reviewer #2: I Don't Know

Reviewer #3: No

3. Have the authors made all data underlying the findings in their manuscript fully available?

Reviewer #1: Yes

Reviewer #2: Yes

Reviewer #3: Yes

4. Is the manuscript presented in an intelligible fashion and written in standard English?

Reviewer #1: Yes

Reviewer #2: Yes

Reviewer #3: Yes

5. Review Comments to the Author

Reviewer #1: This paper is delightful. As the authors emphasize, the work is essential, fills a relevant gap, and addresses a key issue empirically in ways few other studies have. The Introduction motivates the analysis superbly. The consistent reporting of effect sizes is most welcome. The Discussion underscores the essential take-home messages well while touching on the necessary limitations of the current data. I’m trying to take the perspective of the folks who recently asserted, in so many words, that we really just need to keep folks safe during the acute effects and that’ll do the trick. I think that one statistical issue will be critical. I’ve identified a few points that might generate more criticism than the authors want as well.

1. Readers are bound to wonder if a handful of outliers are driving the whole effect. I don’t need a Florence Nightingale Rose or Violin Plot, but the authors should report the skew for anything where they use a Pearson’s or t-test. Anything over an absolute value of 1 will probably need a transformation or a non-parametric alternative or some other strategy to sidestep the issue. (With an N this small, the statistical tests for violations of normality will never reach significance, but they’re ass backwards anyway.) Detailed rationale and potential interventions or transformations appear in:

Osborne JW. Best practices in data cleaning: A complete guide to everything you need to do before and after collecting your data. Sage publications; 2012 Jan 10.

I would guess that anchoring at 1 and taking a log or square-root would not seem to abstruse for most readers.

I’ve got shpilkes about a few other issues, but in PLOS, these potential changes are really the authors’ call:

2. The writing is completely on par with the quality in our field—a low bar. If the authors would consider eschewing passive constructions and grammatical expletives, readability would improve. The word “this” without a definite antecedent, also interrupts the flow. I am guessing that the published version will generate more attention with some tightening. These steps alone would make the Abstract more accessible and provide space to show more details or enthusiasm. Take this paragraph from the Discussion:

Although the model outlined above is well supported by our findings, there is also evidence for reverse causality (i.e., that the acute psychedelic experience enhances the therapeutic alliance). This could be explained by the findings that psilocybin enhances emotional empathy [61] and feelings of connectedness [55,62], and reduces subjective and neurological responses to social exclusion [63]. Furthermore, there is evidence that these effects persist well beyond the acute drug effect [64], and may lead to more secure attachment patterns [60]. In our analysis, peak ratings of acute effects were correlated with a stronger alliance reported one week following the second psilocybin session. Although our analysis cannot reveal causality in this direction, Murphy et al., (2022) were able to demonstrate that the strength of emotional breakthrough in the first psilocybin session predicted improvements in the alliance reported prior to the second session. Taken together, this evidence suggests that psilocybin may work to directly enhance the alliance reported by participants, both acutely and over the longer time course of an intervention.

Perhaps this version reads easier:

Our findings strongly support the model outlined above, showing evidence for reverse causality—specifically, the acute psychedelic experience enhances the therapeutic alliance. Psilocybin's ability to enhance emotional empathy and feelings of connectedness, along with its reduction of subjective and neurological responses to social exclusion, might explain this phenomenon. These effects not only persist but may also foster more secure attachment patterns. In our study, stronger alliances one week after the second psilocybin session correlated with higher peak ratings of acute effects. Murphy et al. (2022) also found that stronger emotional breakthroughs in the first psilocybin session predicted better alliances before the second session. Collectively, this result suggests that psilocybin could directly and progressively enhance the therapeutic alliance.

See what I mean? I think it’s 40 words shorter. Reading it aloud is more pleasant.

3. INTRO. The single-item indicator is bound to generate some concern and skepticism. The authors might want to mention this paper or comparable ones supporting the validity of single-item measures.

Dollinger, S. J., & Malmquist, D. (2009). Reliability and validity of single-item self-reports: with special relevance to college students' alcohol use, religiosity, study, and social life. The Journal of General Psychology, 136(3), 231-242.

4. INTRO The authors might want to mention these publications, some of which might not have appeared at the time of submission, for the sake of comprehensiveness.

Gramling, R., Bennett, E., Curtis, K., Richards, W., Rizzo, D. M., Arnoldy, F., ... & Agrawal, M. (2023). Developing a Direct Observation Measure of Therapeutic Connection in Psilocybin-Assisted Therapy: A Feasibility Study. Journal of Palliative Medicine.

As the title emphasizes, this paper points out a way to sidestep one of the shortcomings of the current data.

Kamilar-Britt P, Gordis EB, Earleywine M. The Therapeutic Alliance in Psychedelic-Assisted Psychotherapy: A Novel Target for Research and Interventions. Psychedelic Medicine. 2023 Aug 18.

This paper also shows the need for multiple raters on the alliance measures.

5. The writing is completely comparable to published work in our field—a very low bar. Changing passive constructions to active voice and eliminating grammatical expletives would improve readability. This paragraph from the Discussion might prove illustrative:

Although the model outlined above is well supported by our findings, there is also evidence for reverse causality (i.e., that the acute psychedelic experience enhances the therapeutic alliance). This could be explained by the findings that psilocybin enhances emotional empathy [61] and feelings of connectedness [55,62], and reduces subjective and neurological responses to social exclusion [63]. Furthermore, there is evidence that these effects persist well beyond the acute drug effect [64], and may lead to more secure attachment patterns [60]. In our analysis, peak ratings of acute effects were correlated with a stronger alliance reported one week following the second psilocybin session. Although our analysis cannot reveal causality in this direction, Murphy et al., (2022) were able to demonstrate that the strength of emotional breakthrough in the first psilocybin session predicted improvements in the alliance reported prior to the second session. Taken together, this evidence suggests that psilocybin may work to directly enhance the alliance reported by participants, both acutely and over the longer time course of an intervention. (169 words)

Our findings strongly support the model outlined above, showing evidence for reverse causality—specifically, the acute psychedelic experience enhances the therapeutic alliance. Psilocybin's ability to enhance emotional empathy and feelings of connectedness, as well as to reduce subjective and neurological responses to social exclusion, explains this relationship. These effects not only persist well beyond the acute drug effect but also potentially foster more secure attachment patterns. Our analysis correlates peak ratings of acute effects with a stronger alliance reported one week after the second psilocybin session. Murphy et al., (2022) demonstrated that the strength of emotional breakthrough in the first psilocybin session predicted improvements in the alliance before the second session. This evidence collectively suggests that psilocybin directly enhances the alliance that participants report, both in the short term and over the course of the intervention. (136 words)

6. METHODS- McDonald’s Omega has become the standard on internal consistency now, in part because the approach requires fewer assumptions. I’m pretty sure SPSS has the option in a straightforward way. Details appear in:

Hayes AF, Coutts JJ. Use omega rather than Cronbach’s alpha for estimating reliability. But…. Communication Methods and Measures. 2020 Jan 2;14(1):1-24.

These authors walk through the issue related to assumptions of tau-equivalence, not that the current paper needs those details.

7. RESULTS- I’m not enjoying the “small,” “medium,” and “large” designations with the effect sizes in the tables. Cohen would admit that he literally pulled the distinctions from thin air and the significance designations certainly help get the point across.

8. DISCUSSION- I think the expression “reverse causality” is ill-advised here. More readers will grow confused than entertained or enlightened.

Reviewer #2: Important note: This review pertains only to ‘statistical aspects’ of the study and so ‘clinical aspects’ [like medical importance, relevance of the study, ‘clinical significance and implication(s)’ of the whole study, etc.] are to be evaluated [should be assessed] separately/independently. Further please note that any ‘statistical review’ is generally done under the assumption that (such) study specific methodological [as well as execution] issues are perfectly taken care of by the investigator(s). This review is not an exception to that and so does not cover clinical aspects {however, seldom comments are made only if those issues are intimately / scientifically related & intermingle with ‘statistical aspects’ of the study}. Agreed that ‘statistical methods’ are used as just tools here, however, they are vital part of methodology [and so should be given due importance]. I look at the manuscript in/with statistical view point, other reviewer(s) look(s) at it with different angle so that in totality the review is very comprehensive. However, there should be efforts from authors side to improve (may be by taking clues from reviewer’s comments). Therefore, please do not limit the revision only (with respect) to comments made here.

COMMENTS: I note your ABSTRACT is well drafted (in my opinion), but is ‘assay type’. It is preferable [refer to item 1b of CONSORT checklist 2010: Structured summary of trial design, methods, results, and conclusions] to divide the ABSTRACT with small sections like ‘Objective(s)’, ‘Methods’, ‘Results’, ‘Conclusions’, etc. which is an accepted practice of most of the good/standard journals [including this one, though ‘The PLoS One Guidelines to Authors’ did not specify an Abstract format, it is desirable]. It will definitely be more informative then, I guess, whatever the article type may be.

Although your article is classified as “Article Type: Research Article” you described the study as “randomized, waiting list-controlled clinical trial” (lines 44-45). Moreover, you have not given/discussed ‘How the required minimum sample size for this study was determined’ which nevertheless is a very-very important question [one of the important items in CONSORT checklist, item 7a] for any type of study (clinical trial or else). This point needs to be discussed in adequate details {including assumptions made at the time of estimation, power (confidence/accuracy/precision in case of single-arm/group studies) of the study, software used, etc.}. Even if this is a sort of off-shoot study of some main (published) study [Effects of Psilocybin-Assisted Therapy on Major Depressive Disorder. JAMA Psychiatry. 2021 May;78(5):1–9], this one being an independent publication, you are expected to cover these details {strangely, such details are not even covered in that paper as well}.

Further though you have clarified/admitted (lines 261-62: It is possible that, given the small sample size, these correlations were influenced by outliers in our sample), one should keep in mind that highly significant (large) value of ‘Pearson’s correlation coefficient’ alone does not imply cause-effect relationship. There are other certain criteria which are to be considered before making any causal inference(s). As ‘P-value’ heavily depends on sample size, it is customary to use the (available in most text books on ‘Biostatistics’ or on ‘www/net’) guidelines [very strongly suggesting] to consider an absolute value of ‘Correlation coefficient’ for interpreting positive or negative correlations (and do not rely only on corresponding ‘P’-value but also consider an absolute value of ‘Correlation coefficient’). [This argument is equally applicable to non-parametric Spearman’s ‘Correlation coefficient (ρ)’ as well.]

As you may already know that “all ‘Clinical Trials’ must follow CONSORT guidelines”. Since you described the study as ‘Clinical Trial’, you are supposed to cover these items in the report. Other important items (other than randomization) are ‘How sample size was determined (Item 7a), Allocation concealment (Item 9), Blinding (Item 11a)} of/in CONSORT checklist which are not adequately described/found in the manuscript. An important word ‘CONSORT’ itself does not appear.

I am sorry to note that my search on WWW/NET reveals that “psilocybin-assisted therapy (PAT) is currently not a standard medical practice – in many parts of the world, it is considered illegal” [update on 07-Jul-2022]. If that is true, my question is “how does it matter 1. if the therapeutic alliance between study participants and intervention facilitators in a psilocybin-assisted therapy (PAT) or 2. if there are relationships between alliance, acute psilocybin experiences, and depression outcomes?”. Unfortunately, I do not understand medical/clinical implications of it but further my search reveals that “Psilocybin is a hallucinogenic chemical found in certain types of mushrooms (sometimes referred to as magic mushrooms). Psilocybin has been shown to produce feelings of euphoria and sensory distortions that are similar to those produced by hallucinogenic drugs like LSD {which is considered to be a very strong hallucinogen}” and that is not desirable, in my knowledge. However, there are/were few very encouraging evidences/studies [example: BMJ Open. 2021; 11(12): e056091. Published online 2021 Dec 1. doi: 10.1136/bmjopen-2021-056091]. No definite answer/conclusion. But any case, authors should adequately discuss this point. Some discussion appears in ‘Introduction’ section (example: lines 85-91), however, that may not suffice.

Please note that, although the measures/tools used are [seems to be] appropriate {like Working Alliance Inventory-Short Revised (WAI-SR), GRID-HAMD, Mystical Experience Questionnaire (MEQ30)}, most of them are likely to yield data that are in ‘ordinal’ level of measurement [and not in ratio level of measurement for sure {as the score two times higher does not indicate presence of that parameter/phenomenon as double (for example, a Visual Analogue Scales VAS score or say ‘depression’ score)}]. Then application of suitable non-parametric (or distribution free) test(s) is/are indicated/advisable [even if distribution may be ‘Gaussian’ (also called ‘normal’)]. Use Mann-Whitney test instead of unpaired ‘t’ test, Spearman’s Correlation Coefficient instead of Pearson’s Correlation Coefficient [even if you get the same/similar results, you are expected to use a right/correct/indicated technique]. Agreed that there is/are no non-parametric test(s)/technique(s) available to be used as alternative in all situation(s), but should be used whenever/wherever they are available. Therefore, in short use suitable non-parametric test(s)/technique(s) while dealing with data that are in ‘ordinal’ level of measurement even if [despite that] the distribution may be ‘Gaussian’. Testing ‘normality’ in sample [by using any normality test(s)} is not required/desired while dealing with data that are in ‘ordinal’ level of measurement [as most of the normality tests are not valid for ‘ordinal’ data].

Though few limitations of the study are mentioned/listed in a section ‘Limitations’ [lines 396 onwards], one very important limitation (namely small sample size) is surprisingly missing. As pointed out in ‘important note’ above “This review pertains only to ‘statistical aspects’ of the study and so ‘clinical aspects’ should be assessed separately/independently [one should carefully consider/look at the clinical implications of the study]. In my opinion, to make this article acceptable (which is quite possible and easy), some amount of re-vision (re-drafting) may be needed. However, please do not limit the revision only (with respect) to comments made here. More improvement is expected. The respected ‘Editor’ may consider accepting/further processing only if found ‘clinical implications’ valuable [i.e., add(s) to clinical knowledge or positively influence clinical practice]. ‘Major revision’ is recommended.

Reviewer #3: The authors provide a nice analysis of human depression scores that emphasizes a relationship between environmental factors (therapeutic alliance) and drug effects (psilocybin-assisted therapy) in clinical population.

My only concerns are the small sample size and the reliance on parametric statistical analyses to assess meaningful relationships:

1. I suggest bolstering the current piece by providing more details to the Methods Section. Specifically, the design of this study, the rationale for the number of subjects selected, and the tools/software used to generate planned analyses should include more details.

2. The authors should carefully consider alternatives to the current use of parametric analyses where the n is small and the range of values is large.

6. PLOS authors have the option to publish the peer review history of their article (what does this mean?). If published, this will include your full peer review and any attached files.

Reviewer #1: No

Reviewer #2: No

Reviewer #3: No

---

## [Author Response · Author response to Decision Letter 0]

17 Jan 2024

Reviewer #1: 

This paper is delightful. As the authors emphasize, the work is essential, fills a relevant gap, and addresses a key issue empirically in ways few other studies have. The Introduction motivates the analysis superbly. The consistent reporting of effect sizes is most welcome. The Discussion underscores the essential take-home messages well while touching on the necessary limitations of the current data. I’m trying to take the perspective of the folks who recently asserted, in so many words, that we really just need to keep folks safe during the acute effects and that’ll do the trick. I think that one statistical issue will be critical. I’ve identified a few points that might generate more criticism than the authors want as well.

1. Readers are bound to wonder if a handful of outliers are driving the whole effect. I don’t need a Florence Nightingale Rose or Violin Plot, but the authors should report the skew for anything where they use a Pearson’s or t-test. Anything over an absolute value of 1 will probably need a transformation or a non-parametric alternative or some other strategy to sidestep the issue. (With an N this small, the statistical tests for violations of normality will never reach significance, but they’re ass backwards anyway.) Detailed rationale and potential interventions or transformations appear in:

Osborne JW. Best practices in data cleaning: A complete guide to everything you need to do before and after collecting your data. Sage publications; 2012 Jan 10.

I would guess that anchoring at 1 and taking a log or square-root would not seem to abstruse for most readers.

Author Response: We thank the reviewer for this suggestion. Given the evidence that shows that Pearson correlations and t-tests are robust against violations of the normal distribution of scores and given that this study is a secondary data analysis from one very small clinical trial which is meant to be exploratory and hypothesis-generating (and not a definitive answer to this research question), we have chosen not to change our analytic method. However, we added details about this rationale to the methods section (lines 186-190) and we added this as a limitation of this study in the discussion section (lines 865-876).

2. The writing is completely on par with the quality in our field—a low bar. If the authors would consider eschewing passive constructions and grammatical expletives, readability would improve. The word “this” without a definite antecedent, also interrupts the flow. I am guessing that the published version will generate more attention with some tightening. These steps alone would make the Abstract more accessible and provide space to show more details or enthusiasm. 

Author Response: We appreciate the stylistic guidance and agree that readability could be improved. As such, we have attempted to remove passive constructions and grammatical expletives throughout the manuscript and to cut words for clarity and to improve flow where possible. In service of this goal, we also replaced the term “psychedelic-assisted psychotherapy” with PAT throughout the manuscript. We also included those recommended edits in Paragraph 7 of the Discussion (lines 806-816), as we believe it significantly enhances readability. 

 Introduction 

3. The single-item indicator is bound to generate some concern and skepticism. The authors might want to mention this paper or comparable ones supporting the validity of single-item measures.

Dollinger, S. J., & Malmquist, D. (2009). Reliability and validity of single-item self-reports: with special relevance to college students' alcohol use, religiosity, study, and social life. The Journal of General Psychology, 136(3), 231-242.

Author Response: We agree that the single-item indicator is worth addressing in more detail. We included an explanation/justification in both the Methods (Lines 179-181) and the Limitations section (Lines 825-845) using the reference you provided above. We also added two additional references (refs. 74, 75) highlighting recently validated (since this study) insight scales for psychedelic therapy. 

4. The authors might want to mention these publications, some of which might not have appeared at the time of submission, for the sake of comprehensiveness.

Gramling, R., Bennett, E., Curtis, K., Richards, W., Rizzo, D. M., Arnoldy, F., ... & Agrawal, M. (2023). Developing a Direct Observation Measure of Therapeutic Connection in Psilocybin-Assisted Therapy: A Feasibility Study. Journal of Palliative Medicine.

As the title emphasizes, this paper points out a way to sidestep one of the shortcomings of the current data.

Kamilar-Britt P, Gordis EB, Earleywine M. The Therapeutic Alliance in Psychedelic-Assisted Psychotherapy: A Novel Target for Research and Interventions. Psychedelic Medicine. 2023 Aug 18.

This paper also shows the need for multiple raters on the alliance measures.

Author Response: Thank you for bringing our attention to these highly relevant recent publications. We felt that they were most appropriately incorporated into the Discussion and Limitations sections. We added the Gramling et al. 2023 citation in our discussion of measuring and defining the therapeutic alliance in psychedelic assisted therapy (Lines 809-814). We added the Kamilar-Britt et al. 2023 paper in the Limitations section (Lines 855-857), suggesting future studies incorporate multiple raters (including third-party). 

5. The writing is completely comparable to published work in our field—a very low bar. Changing passive constructions to active voice and eliminating grammatical expletives would improve readability. 

Author Response: This has been incorporated throughout as mentioned above. 

Methods

6. McDonald’s Omega has become the standard on internal consistency now, in part because the approach requires fewer assumptions. I’m pretty sure SPSS has the option in a straightforward way. Details appear in:

Hayes AF, Coutts JJ. Use omega rather than Cronbach’s alpha for estimating reliability. But…. Communication Methods and Measures. 2020 Jan 2;14(1):1-24.

These authors walk through the issue related to assumptions of tau-equivalence, not that the current paper needs those details.

Author Response: Thank you for this suggestion. However, this statistic has not become the standard in our discipline, and we believe that the alpha value is a better way to communicate the internal consistency of our scales. 

Results 

7. I’m not enjoying the “small,” “medium,” and “large” designations with the effect sizes in the tables. Cohen would admit that he literally pulled the distinctions from thin air and the significance designations certainly help get the point across.

Author Response: Thank you for your comment. Although, possibly arbitrary, the ranking of effect sizes is still widely accepted in the field and offers useful context to readers who are unfamiliar with statistics. That said, we agree that including these distinctions within the table is unnecessary and distracting. Therefore, we have removed them from the table and maintained the footnote to aid in interpretation.

Discussion

8. I think the expression “reverse causality” is ill-advised here. More readers will grow confused than entertained or enlightened

Author Response: We removed this term in favor of a more clear and concise statement (line 697-698): “Our findings also strongly suggest that the acute psychedelic experience directly enhances the therapeutic alliance.”

Reviewer #3: 

Comment: The authors provide a nice analysis of human depression scores that emphasizes a relationship between environmental factors (therapeutic alliance) and drug effects (psilocybin-assisted therapy) in clinical population. My only concerns are the small sample size and the reliance on parametric statistical analyses to assess meaningful relationships:

1. I suggest bolstering the current piece by providing more details to the Methods Section. Specifically, the design of this study, the rationale for the number of subjects selected, and the tools/software used to generate planned analyses should include more details.

Author Response: We have added details to the methods section, including the design of the study and the rationale for choosing the sample size (lines 124-130). We have provided the software that we used (SPSS) in the analysis description. There were no other tools/software used in this analysis. 

2. The authors should carefully consider alternatives to the current use of parametric analyses where the n is small and the range of values is large.

Author Response: We have addressed a similar concern provided by reviewer #1. Given the evidence that shows that Pearson correlations and t-tests are robust against violations of the normal distribution of scores and given that this study is a secondary data analysis from one very small clinical trial which is meant to be exploratory and hypothesis-generating (and not a definitive answer to this research question), we have chosen not to change our analytic method. However, we added details about this rationale to the methods section and we added this as a limitation of this study in the discussion section.

---

## [Decision Letter · Decision Letter 1]

29 Feb 2024

The therapeutic alliance between study participants and intervention facilitators is associated with acute effects and clinical outcomes in a psilocybin-assisted therapy trial for Major Depressive Disorder

PONE-D-23-21083R1

Dear Dr. Davis,

We’re pleased to inform you that your manuscript has been judged scientifically suitable for publication and will be formally accepted for publication once it meets all outstanding technical requirements. We appreciate your careful consideration of the initial feedback by Reviewer 1 and myself. I am in line with your revisions & rationale for the originally chosen use of statistics provided to make assertions about the relationships between Therapeutic Alliance and Mystical *or *Insight Effects. Your study reads well and provides insightful considerations for a burgeoning therapeutic intervention.

Kind regards,

Herb Covington, Ph.D.

Academic Editor

PLOS ONE
---

## [Editor Report · Acceptance letter]

5 Mar 2024

PONE-D-23-21083R1 

PLOS ONE

Dear Dr. Davis, 

I'm pleased to inform you that your manuscript has been deemed suitable for publication in PLOS ONE. Congratulations! Your manuscript is now being handed over to our production team.

Kind regards, 

on behalf of

Dr. Herb Covington 

Academic Editor

PLOS ONE